# Modeling and Prediction of Regular Ionospheric Variations and Deterministic Anomalies

**Mahmoud Rajabi [1,2,*], Alireza Amiri-Simkooei [2,3], Hossein Nahavandchi [1]  and Vahab Nafisi [2]**

1   Department of Civil and Environmental Engineering, Norwegian University of Science and Technology NTNU, 7491 Trondheim, Norway; hossein.nahavandchi@ntnu.no
2   Department of Geomatics Engineering, University of Isfahan, Hezar-Jarib Avenue 81746, Isfahan 73441, Iran; a.amirisimkooei@tudelft.nl (A.A.-S.); nafisi@eng.ui.ac.ir (V.N.)
3   Delft Institute of Earth Observation and Space Systems, (DEOS), Faculty of Aerospace Engineering, Delft University of Technology, Kluyverweg 1, 2629 HS Delft, The Netherlands
*   Correspondence: mahmoud.rajabi@ntnu.no; Tel.: +47-92332254

**Abstract:** Knowledge on the ionospheric total electron content (TEC) and its prediction are of great practical importance and engineering relevance in many scientific disciplines. We investigate regular ionospheric anomalies and TEC prediction by applying the least squares harmonic estimation (LS-HE) technique to a 15 year time series of the vertical TEC (VTEC) from 1998 to 2014. We first detected a few new regular and modulated signals in the TEC time series. The multivariate analysis of the time series indicates that there are diurnal, annual, 11 year, and 27 day periodic signals, as well as their higher harmonics. We also found periods matching with the global positioning system (GPS) draconitic year in the TEC time series. The results from the modulated harmonic analysis indicate that there exists a set of peaks with periods of $1 \pm 0.0027j$ $(j = 1, \dots, 5)$ and $1 \pm 0.00025j$ $(j = 1, 2, 3)$ days. The same situation holds also true for the harmonics higher than the diurnal signal. A model is then adopted based on the discovered periods. This model, which consists of pure and modulated harmonic functions, is shown to be appropriate for assessing the regular variations and ionospheric anomalies. There is a clear maximum TEC at around 22:00 h, which we called the "evening anomaly". The evening anomaly occurs in the winter and autumn, and is dependent on the solar activities. Also, the Semiannual, Winter, and Equatorial anomalies were investigated. Finally, to investigate the performance of the derived model, the TEC values have been predicted monthly, and the results show that the modulated signals can significantly contribute to obtaining superior prediction results. Compared with the pure signals, the modulated signals can improve a yearly average root mean squared error (RMSE) value in the lower and higher solar activities by 20% and 15%, respectively.

**Keywords:** ionospheric anomalies; Least Squares Harmonic Estimation (LS-HE); TEC prediction; TEC time-series

## 1. Introduction

The ionosphere plays an important role in the disciplines of many atmospheric sciences, including telecommunication [1] and global navigation satellite system (GNSS) signals [2]. This layer of atmosphere extends from about 60 km to 2000 km at the top of the Earth's atmosphere, and its main portion is placed between 300 and 400 km [3,4]. The ionosphere is filled with charged particles of both free electrons and ions. Its main layers are known as D, E, F1, and F2 [5]. The total electron content (TEC) is a valuable descriptive quantity for the ionosphere, and is defined as the line integral of the electron density of a column through the ionosphere. It is measured in TECu units, of which one TECu is equivalent to $10^{16}$ electron/m$^2$ [6].

Modeling and prediction of the ionosphere are important issues, on which considerable attention has been focused over the past few decades. The ionospheric effect is one of the most important sources of error in real-time GNSS implementations. Essentially, single-frequency receivers need ionospheric models to eliminate/reduce their signal delays. The International GNSS Service (IGS) final global ionospheric map (GIM) products are released with a time delay of approximately 2 weeks, limiting its applications in real-time. For example, real-time precise point positioning (PPP) and other satellite missions, such a soil moisture and ocean salinity (SMOS) [7] and spaceborne synthetic aperture radar (SAR) [8] require ionospheric information in real time. Therefore, modeling and prediction of global vertical TEC (VTEC) are essential. Many researchers have attempted to investigate the ionospheric variation by using several forecasting models and mapping algorithms, such as least squares, neural networks, kriging, empirical orthogonal functions, and autoregressive moving average model (ARMA) [7,9–16]. Most of these methods have focused on the study of global and regional parts of the ionosphere. For example, Gong and Dang [17], using IGS TEC global data and the improved analysis of variance method for short-term forecasting, showed the root mean squared error (RMSE) distribution of forecasting values corresponding to each set of data is between 1.5-2.5 TECu, and they improved forecasting by the weighted method.

Ionospheric variations can generally be categorized as regular and irregular [18]. The regular parts happen almost in cycles because of the Earth's rotation and revolution and sun-related features, such as 11 year and 27 day variations originated from sunspot variations [5]. Therefore, they can be modeled with considerable accuracy. Such variations include changes due to diurnal, annual, and solar cycle signals, to name a few. The irregular parts, however, are mainly due to abnormal solar behavior, such as Sporadic E, caused by unusual and irregular cloud-like patches of high ionization; sudden ionospheric disturbances, occurring without warning and usually related to solar eruption; and ionospheric storms caused by disturbances in the Earth's magnetic field related to the rotation of the sun and solar eruptions [5,19]. However, modeling of irregular parts of the ionosphere is rather difficult.

The effects in the ionosphere that seem unpredictable at first are called anomalies. There exist different kinds of ionospheric anomalies. For example, the ionosphere has more electron density during the day in winter compared to the summer. This phenomenon is called a "winter" or "seasonal" anomaly [20]. The seasonal changes in NmF2 (maximum electron density of the F2 region) are related to changes of constituents as the summer hemisphere is being heated and the lighter neutral constituents are transferred to the winter hemisphere. Also, the winter anomaly's frequency and area diminish with decreasing solar activity [21]. By investigating TEC data, Jakowski and Förster found that when the sun's activity is not high, there is a nighttime winter anomaly (NWA) effect in mid-latitudes in the American and Asian sectors [22]. Mikhailov et al. [23,24] and Farelo et al. [24] found two peaks in NmF2 night changes. These two peaks occurred before and after midnight, showing different characteristics because of different physical mechanisms. Meza et al. [20] used principal component analysis (PCA) and wavelet transform (WT) for VTEC, and they found that the winter anomaly is recorded at noon near the geomagnetic poles, the effect is more important during high solar activity, and the NWA effect is clear.

There are some other anomalies related to the normal variations of the ionosphere. The semiannual anomaly is caused because the NmF2 is greater at equinoxes than at solstices [25]. Although temperature fluctuations can be the reason for the observed semiannual variations in the height of the F2 peak, changes in NmF2 require additional variations in the neutral composition at lower heights. The semiannual changes happen in the daytime around the world; their occurrence at night is not observed except in South America and near the equator [26,27]. Balan et al. [28] observed an equinoctial asymmetry in the ionospheric peak. They concluded that the NmF2 values in March are larger than those in September by about 50% at all local times. Ma et al. [29] found that the semiannual changes in the diurnal tide in the lower thermosphere lead to changes in the ionospheric equatorial anomaly with a fountain effect. Meza et al. [20] showed that the semiannual anomaly is recorded at noon, mainly

at middle and low latitudes, and has a close relationship with solar activity; at night, this anomaly is recorded during high solar activity. The values of the VTEC at the March equinox exceed that of the September equinox. There also exist variations of electron density in the lower geomagnetic latitudes, reaching a maximum at geomagnetic latitude 15°, called the "equatorial" or "Appleton" anomaly [30]. Although the above-mentioned variations are known as "anomalies", considerable parts of their effects can be modeled as deterministic signals in the functional model [16]. This is shown in the present contribution.

The ionospheric influence on the GNSS signals is dependent on the number of TEC. We aim to investigate regular anomalies and predict TEC values using a model that includes both pure (original wave-like diurnal) and modulated signals, having three components: the original wave and two sinusoidal waves whose frequencies are slightly above and below the original frequency. Since the TEC values have cyclic variations in their regular part, they can consequently be modeled by a series of periodic functions, such as sinusoidal ones. The least squares harmonic estimation (LS-HE), as a method used to analyze frequencies, is applied to the TEC time series derived from ionospheric models. LS-HE has a few unique characteristics. The method is not limited to evenly spaced data or to integer frequencies [15]. The method can also be employed to detect the regular and modulated variations of the ionosphere. The time series in our analysis consists of 15 years of bi-hourly TEC values provided by the Jet Propulsion Laboratory (JPL).

This work is a follow-up to the study by Amiri-Simkooei and Asgari [16]. It differs from that one, however, in the following three aspects: (1) we aim to detect new regular and modulated signals in the TEC time series. A longer time series (now 15 years) allows us to detect new pure (e.g., GPS draconitic year period) and modulated signals in the TEC time series. (2) Using all detected signals, we then investigate the structure and nature of different ionospheric anomalies. (3) We use the final functional model of the TEC time series to predict the ionosphere and investigate the model in local time. The derived functional model is evaluated by predicting TEC time series using (a) pure harmonic signals and (b) both pure and modulated harmonic signals. This is accomplished both in low and high solar activities.

## 2. Methodology

The methodology in the current paper includes two main stages, illustrated in Figure 1 as a methodology flowchart. The first part focuses on the LS-HE application to ionospheric time-series, and the second part shows the model forming and prediction, as explained below.

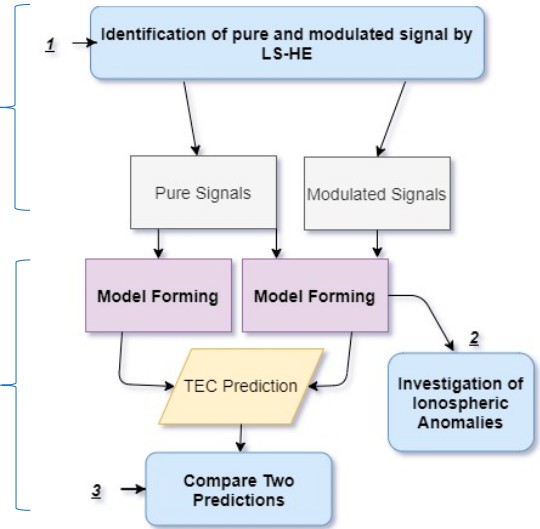

**Figure 1.** Methodology flowchart based on least squares harmonic estimation.

*2.1. Least Squares Harmonic Estimation (LS-HE)*

Most parts of the ionospheric variations, including ionospheric anomalies, are of periodic nature. Therefore, periodic base functions, such as pure and modulated sinusoidal signals, can be used to model and predict the ionosphere. We employ LS-HE to identify and investigate possible pure and modulated signals in TEC time-series. The LS-HE was introduced and applied to the GPS position time series by [31,32]. The LS-HE was also applied to the TEC time series by Amiri-Simkooei and Asgari [16]. We briefly review this method.

Consider the following linear model of observation equations:

$$E(\mathrm{y}) = Ax, \ D(\mathrm{y}) = Q_y \tag{1}$$

where $A$ is the $m \times n$ design matrix, $Q_y$ is the $m \times n$ covariance matrix of the of observations $y$, x is the vector of unknown parameters, $m$ is the number of observations, $n$ is the number of parameters, and $E$ and $D$ are the expectation and dispersion operators, respectively. In the case of a time series, a preliminary functional model $E(\mathrm{y}) = Ax$ may simply include a linear regression model (see [16], for example). Further improvement of the functional model is achieved using LS-HE. The LS-HE formulation has been presented in the univariate (single) and multivariate (multiple) time series. A few features of the LS-HE compared to its counterpart, Fourier spectral analysis, are as follows:

- LS-HE is a generalized form of the Fourier spectral analysis. It is thus neither limited to evenly spaced data nor to integer frequencies. In many real time series, there are some considerable gaps in the data. LS-HE can handle gaps in the data [16,33].
- In the earlier studies by Vaníček [34], Lomb [35], and Scargle [36], a modified variant of Fourier analysis, called least squares spectral analysis, applicable to unevenly spaced data series has been presented. LS-HE is superior over this method because it may, in addition, include the following terms into the analysis: (1) the linear trend $Ax$, as a deterministic part of the model, and (2) the covariance matrix $Q_y$, as a stochastic part of the model [31].
- A unique feature of LS-HE is its multivariate formulation. The performance of the multivariate formulation is superior over its univariate formulation, because it allows the detection of the common-mode signals in a time series. Parts of such signals cannot be detected in the univariate analysis [16,33].
- LS-HE can also be applied to detect modulated signals. This is also another important feature of LS-HE. Many real time series are suspected to have modulated sinusoidal signals rather than pure sine functions. LS-HE can detect possible modulated signals.

We apply the multivariate LS-HE of the multivariate linear model to detect common-mode pure and modulated signals of multiple TEC time series. To combine TEC time series in different longitudes and latitudes ($r$ is the number of time series) Equation (1) is generalized to

$$E(\mathrm{vec}(Y)) = (I_\mathrm{r} \otimes A)\,\mathrm{vec}(x) + (I_\mathrm{r} \otimes A_k)\,\mathrm{vec}(X_k) \tag{2}$$

where vec is the vector operator, $\otimes$ is the Kronecker product, which is an operation on two matrices of arbitrary size resulting in a block matrix, $I$ is the identity matrix, $A$ and $A_k$ are the designs matrix. The structure introduced in $(I_\mathrm{r} \otimes A_k)$ indicates that there exists a common periodic signal in all of the series. The $m \times r$ matrix $Y = [y_1 \ y_2 \ldots y_\mathrm{r}]$ collects observations from the $r$ series, as do the $n \times r$ matrices $X = [x_1 \ x_2 \ldots x_\mathrm{r}]$ and $X_k = [x_{1\mathrm{k}} \ x_{2\mathrm{k}} \ldots x_{\mathrm{rk}}]$ for the unknowns. For a detailed description of the theory and applications of LS-HE, we may refer to Amiri-Simkooei and Asgari [16].

## 2.2. Total Electron Content Modeling and Prediction

The TEC time series consists of some pure and modulated harmonic signals [16]. To model regular variations of the ionosphere, one can form the functional model of the time series with the following two terms:

$$y(t) = y_1(t) + y_2(t) \tag{3}$$

where

$$y_1(t) = \sum_{i=1}^{p} a_i^1 \cos \omega_i t + a_i^2 \sin \omega_i t \tag{4}$$

consists of a series of pure sinusoidal signals expressing the regular pure harmonic term ($\omega$ is angular frequency), and

$$y_2(t) = \sum_{k=1}^{q} b_k^1 \cos \omega_k^s t + b_k^2 \sin \omega_k^s t + b_k^3 \cos \omega_k^d t + b_k^4 \sin \omega_k^d t \tag{5}$$

consists of a few modulated sinusoidal signals expressing the regular modulated harmonics by the frequencies $\omega_K^S = \omega_2 + \omega_1$ and $\omega_K^d = \omega_2 - \omega_1$, with $p$ being the number of pure harmonics and $q$ the number of modulated signals. The combined effect of $y_1(t)$ and $y_2(t)$ is considered to be the entire regular variations of the ionosphere. The coefficients $a_i^1$ and $a_i^2$ and $b_k^i$, $i = 1, 2, 3, 4$ are unknowns. The estimation of these coefficients is referred to as "modeling". After the identification of pure and modulated signals of the TEC time series and forming the model, we aim first to investigate the ionospheric anomalies, and second to predict the TEC values by the identified model. The first part detects the anomalies of the ionosphere, while the second part predicts the behavior of the ionospheric TEC using the estimated coefficients. Further details are explained as follows.

Two scenarios were used for the prediction and identification of the TEC. The first scenario considers only pure signals. The second scenario takes both pure and modulated signals into consideration. For both scenarios, the design matrix $A$ is based on the observation equations in Equations (4) and (5). Least squares estimation of the pure and modulated signal parameters for a given TEC time series is obtained through the following equation:

$$\hat{x} = \left( A^T A \right)^{-1} A^T y \tag{6}$$

where $y$ is the vector of observables, $\hat{x}$ is the vector of estimated parameters, and the design matrix $A$ includes only pure signals under the first scenario and both pure and modulated signals under the second scenario. The pure signal introduces two columns to the design matrix, while a modulated signal introduces four columns.

The identification stage is related to the time instants $t_1, t_2, \ldots t_m$, of which time series observations $y_1, y_2, \ldots y_m$ are available. In the estimation (modeling) stage, the signals parameters are estimated using the observations $y_1, y_2, \ldots y_m$ in the time instants $t_1, t_2, \ldots t_m$. Then the time series can be predicted outside the time-series interval $t_1, t_2, \ldots t_m$—i.e., within $t_{m+1}, t_{m+2} \ldots t_{m+k}$, with $k$ being the total number of time instants for prediction. We may use the corresponding design matrix from the following equation:

$$A_p = \left[ A_1^{(1)}, \ldots, A_1^{(p)}, A_2^{(1)}, \ldots, A_2^{(q)} \right] \tag{7}$$

where $A_1^{(i)}$, $i = 1, 2, \ldots, p$ are the design matrices corresponding to the pure signals;

$$A_1^{(i)} = \begin{bmatrix} \cos \omega_i t_{m+1} & \sin \omega_i t_{m+1} \\ \cos \omega_i t_{m+2} & \sin \omega_i t_{m+2} \\ \vdots & \vdots \\ \cos \omega_i t_{m+k} & \sin \omega_i t_{m+k} \end{bmatrix} \tag{8}$$

and $A_2^{(j)}$, $j = 1, 2, \ldots, q$ are the design matrices corresponding to the modulated signals;

$$
A_2^{(j)} = \begin{bmatrix}
\cos \omega_j^s t_{m+1} & \sin \omega_j^s t_{m+1} & \cos \omega_j^d t_{m+1} & \sin \omega_j^d t_{m+1} \\
\cos \omega_j^s t_{m+2} & \sin \omega_j^s t_{m+2} & \cos \omega_j^d t_{m+2} & \sin \omega_j^d t_{m+2} \\
\vdots & \vdots & \vdots & \vdots \\
\cos \omega_j^s t_{m+k} & \sin \omega_j^s t_{m+k} & \cos \omega_j^d t_{m+k} & \sin \omega_j^d t_{m+k}
\end{bmatrix}
\tag{9}
$$

In the first scenario, only the $A_1^{(i)}$ values are used, whereas in the second scenario, both $A_1^{(i)}$ and $A_2^{(j)}$ values will be used. The predicted (extrapolated) values can then be obtained as follows:

$$
y_p = A_p \hat{x}
\tag{10}
$$

To investigate the performance of the predicted values, a comparison criteria based on the root mean squared error (RMSE) values will be used:

$$
\text{RMSE} = \sqrt{\frac{1}{n} \sum_{i=0}^{n} \left( y_i - \hat{y}_{pi} \right)^2}
\tag{11}
$$

where $y_i$ is the real value and $y_{pi}$ is the predicted value of the VTEC.

## 3. Results and Discussions

### 3.1. Data Set Description

In this study, we used about 15 years of VTEC data from 1998 to 2014, with global coverage provided by JPL. Global ionospheric maps are generated on a bi-hourly basis at JPL using data from over 100 GPS sites of the IGS and other institutions. The vertical TEC is modeled in a solar-geomagnetic reference frame using bi-cubic splines on a spherical grid. A Kalman filter is used to solve simultaneously for instrumental biases and VTEC on the grid (as stochastic parameters) [37,38]. The data is open-access in this file transfer protocol (FTP) server ftp://cddis.gsfc.nasa.gov/gnss/products/ionex [39]. Each JPL GIM VTEC map given in the data center has a spatial resolution of 2.5° in latitude and 5° in longitude, with bi-hourly time intervals covering from 180°W to 180°E and from 87.5°N to 87.5°S. Figure 2 shows an example of a TEC time series generated in this study. The JPL GIM file format is IONEX (The IONosphere Map Exchange Format).

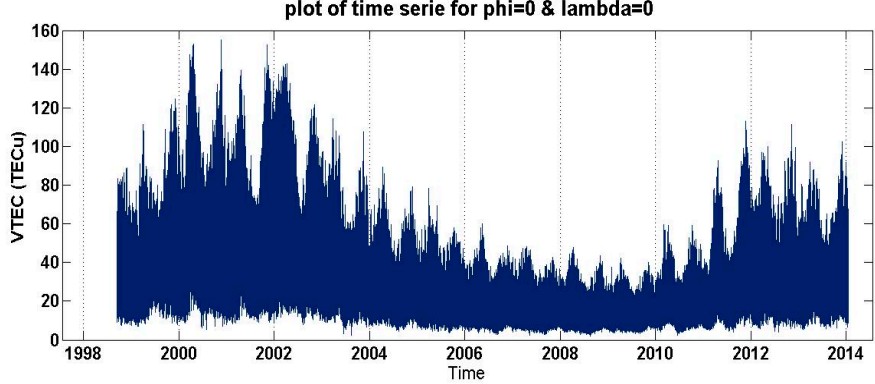

**Figure 2.** One typical example of a total electron content (TEC) time series generated in this study. It was obtained from the global ionospheric maps (GIMs) from 1998 to 2014, on the basis of the bi-hourly time interval.

Many global GIMs are based on spherical harmonic expansion up to a degree and order of 15 or so. The information in such models has a spatial resolution that is not consistent with 2.5° × 5° spacing. However, 2.5° × 5° products were available in the JPL data center, and are used in this study.

### 3.2. Pure Periodic Signals

We applied the multivariate variant of the LS-HE method, presented in Section 2, to the TEC time series in two directions: along the equator and along the prime meridian. This method can combine individual time series to obtain the common-mode, least-squares power spectrum of the time series, and hence their common-mode periods. The lowest frequency used in the estimation is based on the time span of the data (15 years), and is performed to have one cycle over the total time span. The highest frequency is based on the sampling rate, the Nyquist frequency, which is 4 h in this analysis.

Figure 3 illustrates the resulted spectrum for 71 TEC time series located at $\lambda = 0°$ and $]\varphi = [-87.5° : 2.5° : 87.5°]$, i.e., at a specific longitude and different latitudes. It shows that there are regular signals, in agreement with the findings of Amiri-Simkooei and Asgari [16]. Common-mode regular signals can be detected using the multivariate analysis. The spectrum shows a periodic pattern with periods of $24/n$ hour, $n = 1, 2, \ldots, 6$. Therefore, the spectral peaks are at the harmonics of 1–6 cycles per day. Higher harmonics could likely be seen if the sampling rate was higher than 2 h. A periodic pattern with periods of $365.25/n$ days, $n = 1, 2, \ldots, 4$, is the most obvious annual signal and its higher harmonics. In addition, there are three signals with periods of 27 days, 11 years, and 5.5 years, which are in conjunction with the solar cycle periods. As can be seen in Figure 3, the detected signals (peaks) are more elongated/flattened at lower frequencies (e.g., 5.5 year or 11 year periods), which is due to the leakage effect in the spectral analysis [33]. Longer time series are required to see the detected signals sharper.

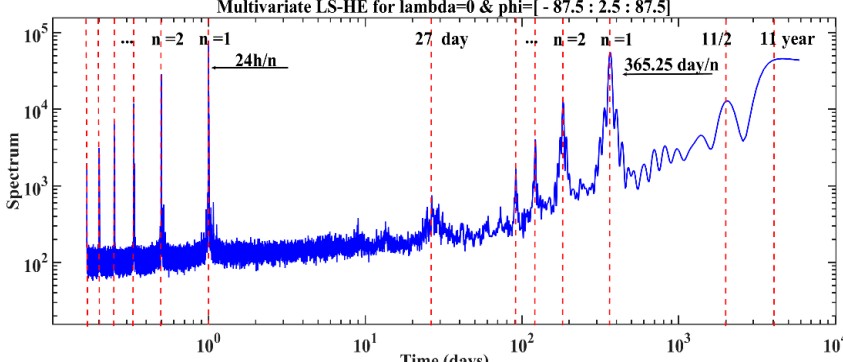

**Figure 3.** The multivariate, least-squares power spectrum of 71 bihourly total electron content (TEC) time series covering latitudes ranging from −87.5 to 87.5 degrees and the period 1998–2014. The vertical dashed lines indicates diurnal and annual signals, along with their higher harmonics, a 27 day signal, an 11 year signal, half of the 11 year signal, etc.

Figure 4 illustrates the power spectrum of 72 TEC time-series located at $]\phi = 0°$ and $\lambda = [-180° : 5° : 180°]$, i.e., at a specific latitude and in different longitudes. We observed similar draconitic signals in the TEC time series as observed in the time series of the GPS coordinates [33]. The GPS satellite's orbital error and multipath are recognized as two possible sources of these signals [40]. The effect of Earth shadow crossing of GPS satellites is another probable reason for these signals [41]. These signals can be seen in the TEC time series from JPL with a period of $351.4/n$ days, where $n = 2, 6, 7, \ldots, 15$ (mainly at the higher frequency of the draconitic effect). As seen in Figure 4, a similar flatness problem can be observed in the lower frequencies portion of this spectrum (e.g., draconitic year period), which occurs because of the same leakage effect mentioned above. This leads to a shift between the annual signal and the draconitic year period. The length of the TEC time series is not yet

long enough to distinguish between the annual signal and the draconitic year period; the time-series should be at least 25 years long [33].

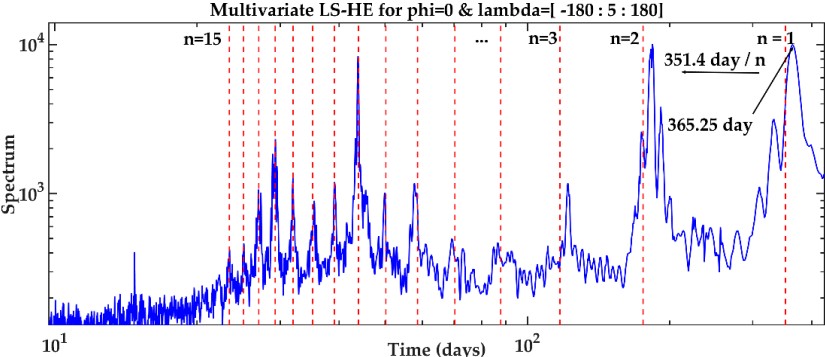

**Figure 4.** The multivariate, least-squares power spectrum of 72 bihourly total electron content (TEC) time series covering longitudes ranging from −180 to 180 degrees and the period 1998–2014. The vertical dashed lines indicate draconitic signals and higher harmonics found in this study.

### 3.3. Modulated Periodic Signals

The regular signals with the periods mentioned above are not the only signals in the TEC time-series. A zoom-in on the diurnal signal in Figure 3 shows signals close to that diurnal signal (Figure 5). There are a series of peaks, with the periods close to the diurnal signal having the periods of $1 \pm j/365.25 = 1 \pm 0.0027j$ days ($j = 1, 2, \ldots, 5$). Each pair of plus and minus signals corresponds to the diurnal signal modulated with the annual signal. The same situation holds also true for harmonics higher than the diurnal signal, i.e., semi-, tri-, and quad-diurnal signals. These are consistent with the findings of Amiri-Simkooei and Asgari [15]. Furthermore, closer to the diurnal signal, we also found other signals with periods of $1 \pm j/(11 \times 365.25) = 1 \pm 0.00025j$ days ($j = 1, 2, 3$), which are modulated into the diurnal signal (Figure 5, vertical dashed lines within the ellipse). The diurnal signal is modulated with 11 year signals of the solar cycle. The same situation holds also true for the higher harmonics of the diurnal signal.

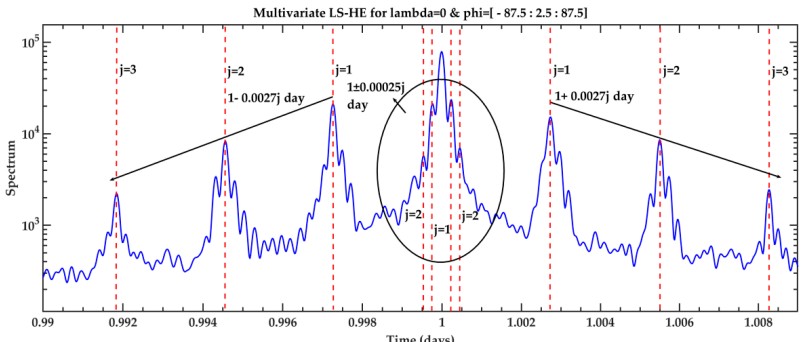

**Figure 5.** Zoom-in of the multivariate least squares spectrum of the total electron content (TEC) time-series in Figure 3. The diurnal signal is modulated with the annual and 11 year signals.

### 3.4. Regular Ionospheric Anomalies

After considering the multivariate analysis of the TEC time series, we developed a model including the above-mentioned periods (24/$n$ hour, 27 day, 365.25/$n$ year, 351.4/$n$ year, $1 \pm 0.0027j$, $1 \pm 0.00025j$, 11 year, and 5.5 year). The ionospheric variations were then modeled to further study the ionospheric anomalies. This goes through the following subsections.

### 3.4.1. Semiannual Anomaly

Figure 6 shows the vertically modeled TEC values at the latitude and longitude of $0°$ in 2012. The vertical TEC values close to the equinoxes (80th and 264th days of a year, equivalent to 21 March and 23 September, respectively) are larger than those for the solstices (172nd and 355th days of the year, equivalent to 21 June and 22 December), in agreement with the previous work. Figure 7 shows the vertical TEC values at the equinoxes and solstices in local times, at different latitudes, and when $\lambda = 0°$. We observe that this anomaly occurs at lower and mid-latitudes during the daytime. Also, the VTEC value of the March equinox is larger than that of the September equinox. Similar results were reported by Millward et al. [27], using the coupled thermosphere ionosphere plasmasphere (CTIP) model; by Balan et al. [28], using the middle and upper atmosphere (MU) radar observations; and by Meza et al. [20], using PCA and WT for VTEC. In addition, we also observed that the March equinox has a similar pattern to the September equinox at different latitudes. This is not the case, however, for the June solstice when compared to the December solstice. An interchange between the southern and northern hemispheres at different latitudes can be observed (Figure 7C,D).

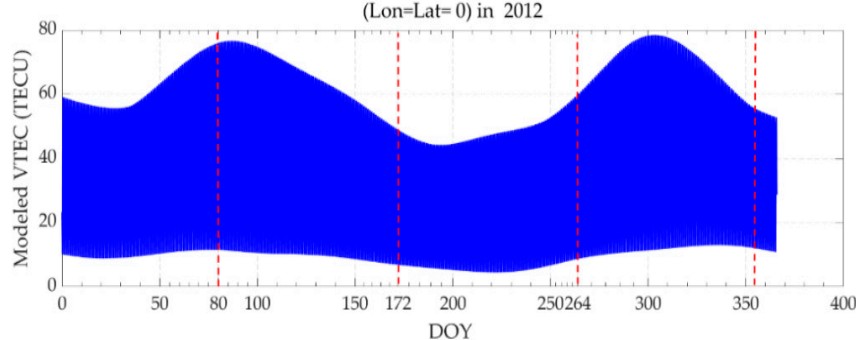

**Figure 6.** Vertically modeled total electron content (TEC) values at the latitude and longitude of $0°$ in 2012; the horizontal axis shows time (day), and the vertical axis shows the modeled vertical total electron content (VTEC) values. Equinoxes and solstices are shown by vertical dashed lines.

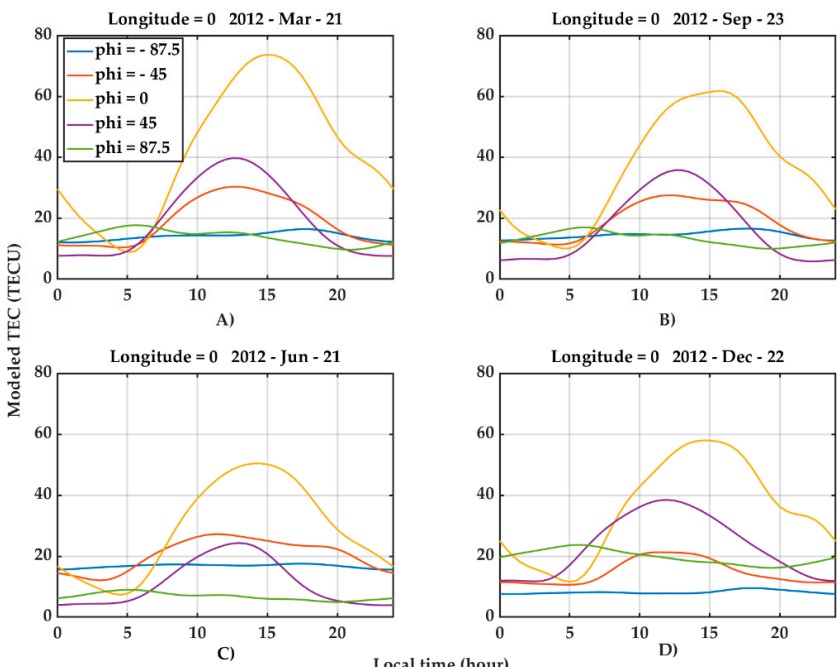

**Figure 7.** Vertical total electron content (VTEC) values of equinoxes and solstices in the local time and at latitude: $±87.5°$, $±45°$, $0°$, and $\lambda = 0°$. (**A,B**) show values in the equinoxes, (**C,D**) show values in the solstices.

### 3.4.2. Seasonal Anomaly

Figure 8 shows the modeled VTEC values from 2010 to 2013, and Figure 9 shows them in the local time for a single day in different seasons and latitudes during high solar activity. We observe that the lowest values of TEC belong to summer, and likely occur in July; in addition, the TEC has the highest values in autumn, which occur in October. This is known as the seasonal or winter anomaly. Figure 10 illustrates the modeled VTEC values in the summer and the winter in 2008 during low solar activity. Comparison of the VTEC values in 2012 (solar maximum, Figure 9) and 2008 (solar minimum, Figure 10) shows that winter anomaly is larger during high solar activities than low solar activities. Also, this anomaly is larger in the daytime than the nighttime. Therefore, this phenomenon occurs mostly in daytime. In addition, there is a nighttime peak in autumn and winter, which is discussed below (evening anomaly). Similar results were achieved by Mikhailov et al. [23], Farelo et al. [24], and Meza et al. [20].

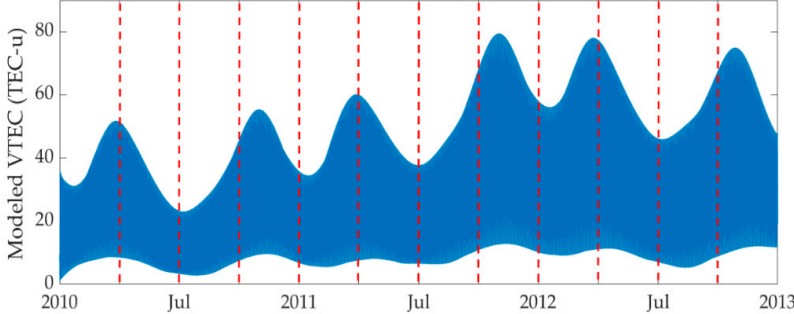

**Figure 8.** Modeled vertical total electron content (VTEC) values from 2010 to 2013 (solar maximum).

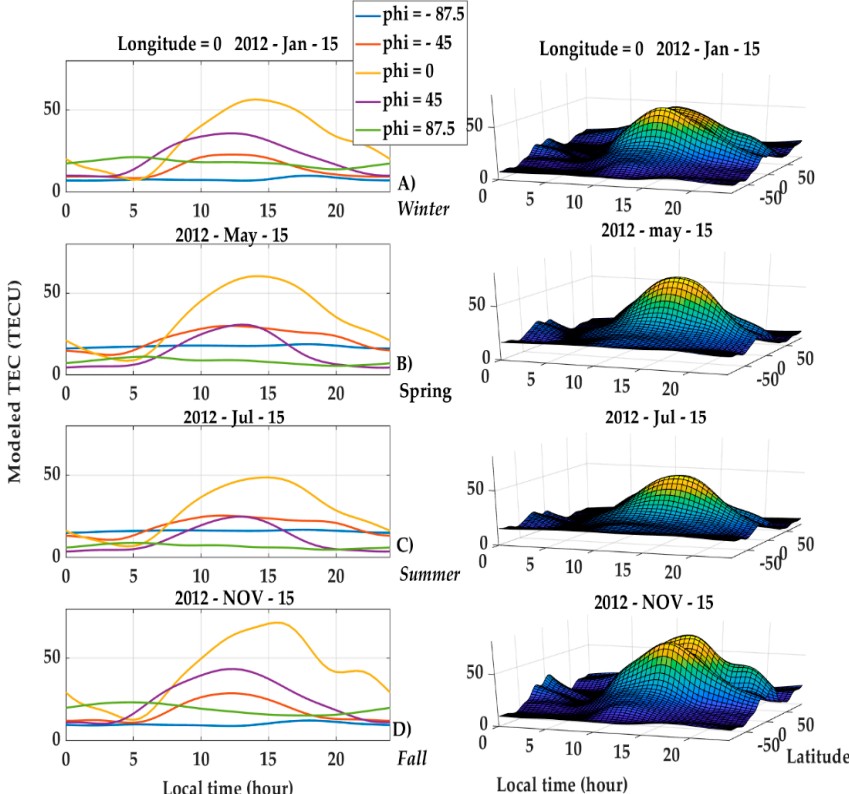

**Figure 9.** Modeled vertical total electron content (VTEC) values in a single day and in different seasons in two and three dimensions, in terms of the local time (the color of lines represent different latitudes).

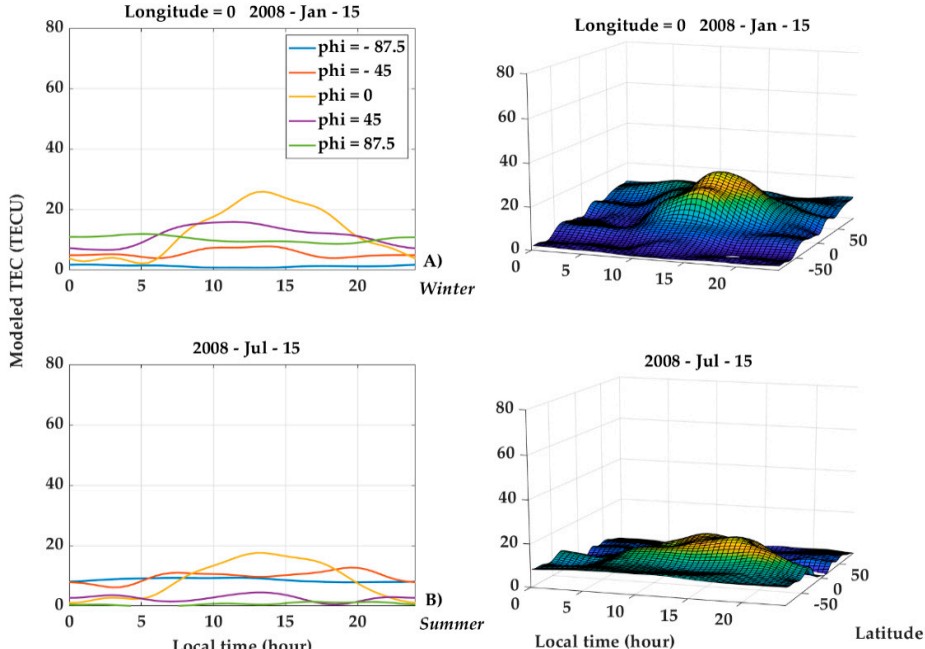

**Figure 10.** Modeled vertical total electron content (VTEC) values on 15 July 2008 (summer) and 15 January 2008 (winter).

### 3.4.3. Evening Anomaly

As can be seen in Figure 9, there is a peak around 22:00 in the autumn and winter. We call this the evening anomaly. This phenomenon is highly dependent on solar activities. They are stronger and longer in high solar activities. This is represented in Figures 11 and 12, where the VTEC values are shown for 2012 (high solar activity) and 2008 (low solar activity) in terms of the local time in different latitudes and when $\lambda = 0°$. The evening anomaly most likely starts to occur in August and reaches its highest value in November. It gradually disappears until the end of winter in equatorial regain (Figures 11 and 12). Moreover, this nocturnal event occurs in equatorial and mid-latitude. This phenomenon can be observed in the mid-region in summer (Figures 9C and 10B). Mikhailov et al. [23], Farelo et al. [24], and Meza et al. [20] found the same phenomenon in winter, known as NWA. The pre-midnight peaks in winter occur due to a heavy, equatorward, thermospheric wind raising the F2 layer to heights with a lower recombination rate [23].

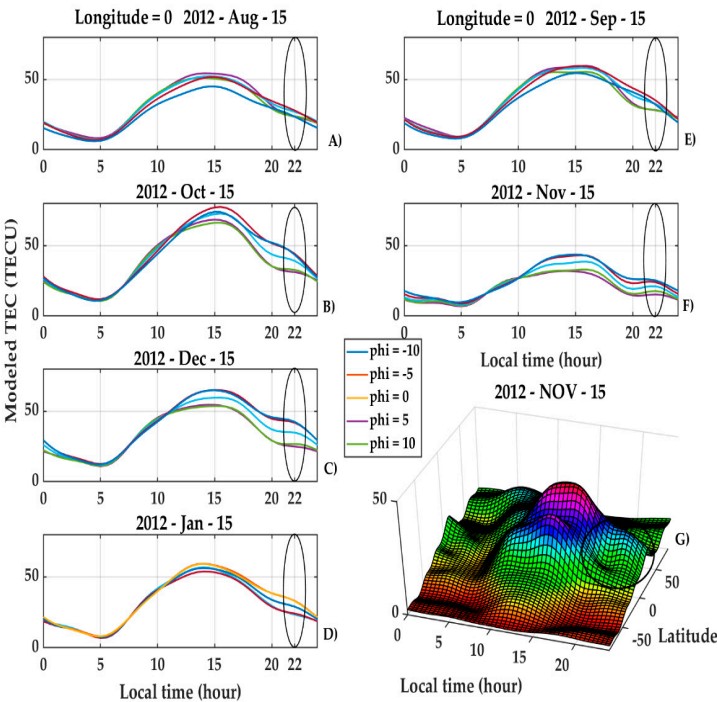

**Figure 11.** Vertical total electron content (VTEC) values in the local time in fall and winter 2012. In the three-dimensional (3D) figure, the values represent the VTEC in local time, in different latitudes, and when $\lambda = 0°$.

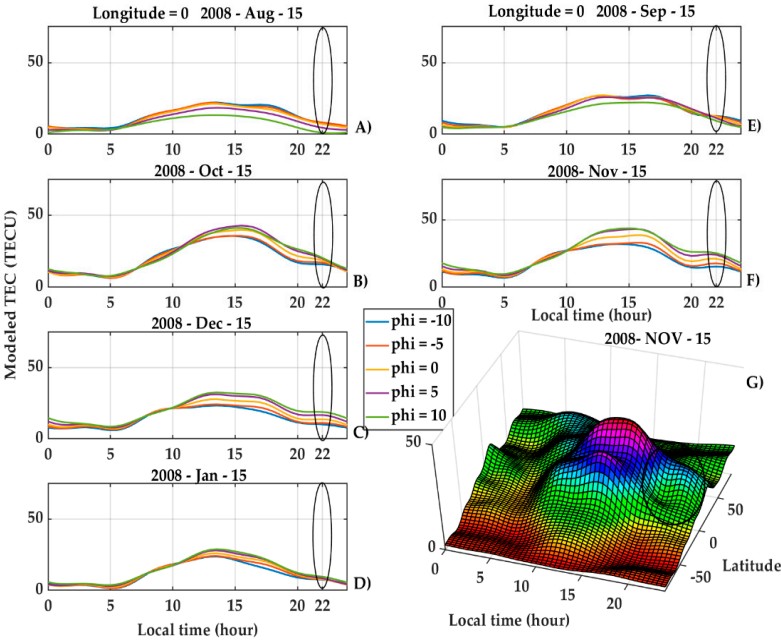

**Figure 12.** Vertical total electron content (VTEC) values in the local time on one day of the month in fall and winter 2008. In the 3D figure, the values represent the VTEC in local time, in different latitudes, and when $\lambda = 0°$.

### 3.4.4. Equatorial Anomaly

Figures 10–12 (three-dimensional figures) indicate that the maximum TEC values occur around latitudes $\pm20°$, peaking at $\pm15°$. This phenomenon develops in the morning at around 10:00, and exists well beyond the sunset. This structure is known as the equatorial or Appleton anomaly. Our results are in agreement with those presented by Schlatter [30].

*3.5. Total Electron Content Prediction*

The proposed method of TEC prediction is based on the extrapolation approach, which requires no input of physical observations for the time interval of prediction. Equations (6) and (10) were used to perform a prediction of VTEC values. Two scenarios were considered. The first one uses only pure signals, whereas the second scenario implies both pure and modulated signals. We investigated whether or not the modulated signals improved the prediction results. The VTEC values were predicted (extrapolation) for each month of 2008 (solar minimum) and 2013 (solar maximum). In this process, to predict the VTEC for each month, we used the data of the past 36 months (3 years) for model fitting. In this way, we could investigate the effect of solar activities and modulated signals on the model performance. The performance of each of the two scenarios is investigated by the RMSE between the real data and the predicted values.

Table 1 shows that the RMSE of the predicted values is higher in high solar activities than in low solar activities, and there is about 5 TECu difference in the monthly mean RMSE values of these two. In other words, TEC values show better prediction in low solar activities. This is, first of all, due to lower TEC values in the solar minimum than in the solar maximum. The errors are also expected to be lower in the solar minimum. The other reason is likely that the model could not sufficiently be adapted to the sudden changes in (high) solar activities. Moreover, a comparison of the two prediction scenarios, based on the pure and the combination of pure and modulated signals, shows a reduction of 3.2 TECu in the monthly RMSE experiences in some of the months. Figure 13 shows the performance of the prediction model in these two scenarios. The dashed lines show the yearly average of the RMSE values. This figure shows that the modulated signals significantly improve the quality of the prediction, reducing the yearly RMSE 3.6 to 2.9 TECu in low solar activities, and 8.8 to 7.5 TECu in high solar activities. In other words, the modulated signals lead to the RMSE reduction of 0.7 TECu and 1.3 TECu in the low and high solar activities, respectively.

**Table 1.** Monthly root mean squared error (RMSE) of vertical total electron content (VTEC) prediction in low and high solar activities, using only pure and both pure and modulated signals at the latitude and longitude of 0°.

| Months | Low Solar Activities (2008) | | High Solar Activities (2013) | |
| :---: | :---: | :---: | :---: | :---: |
| | **Only Pure Signals** | **Both Modulated and Pure Signals** | **Only Pure Signals** | **Both Modulated and Pure Signals** |
| 1 | 3.8 | 3.5 | 7.3 | 7.1 |
| 2 | 2.6 | 3.1 | 5.7 | 4.1 |
| 3 | 3.5 | 2.9 | 9 | 5.6 |
| 4 | 4.5 | 4.3 | 9.9 | 7.5 |
| 5 | 4.5 | 3.9 | 12.7 | 12.4 |
| 6 | 4.3 | 2 | 4.7 | 4.8 |
| 7 | 4.4 | 1.9 | 5.6 | 5.7 |
| 8 | 4.3 | 2.7 | 8.6 | 8.3 |
| 9 | 3 | 2.3 | 16 | 15.7 |
| 10 | 3.7 | 3.9 | 9.6 | 6.7 |
| 11 | 2.4 | 2.2 | 10 | 6.8 |
| 12 | 2.4 | 2.2 | 6.4 | 5.5 |
| **Mean** | **3.6** | **2.9** | **8.8** | **7.5** |

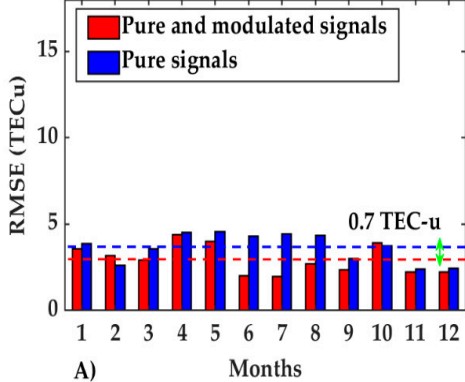 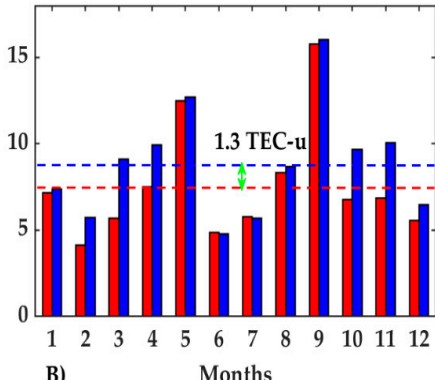

**Figure 13.** Root mean squared error (RMSE) values for total electron content (TEC) prediction in (**A**) low (2008) and (**B**) high (2013) solar activities at the latitude and longitude of 0°. The red color is related to the model with both pure and modulated signals, and the blue color is related to the model only with pure signals. The dashed line shows the yearly RMSE.

There is some ongoing research in the field of TEC forecasting. The study conducted by Gong and Dang [16] using IGS TEC (global) data showed the RMSE distribution of forecasting value corresponding to each set of data is between 1.5-2.5 TECu. Huang and Yuan [41] use the radial basis function (RBF) neural network to forecast ionospheric 30 min TEC. Their predicted results have an RMSE of less than 5 TECu [42]. Liu and Chen, for the regional long-term interval, use spherical cap harmonic analysis, and results of prediction have 4.5 TECu for a 2 month latency [13]. Compared with our results, one may argue that they provide superior results. Note, however, that we used long-term prediction, and the area of investigation was the most complex equatorial region. These all indicate the importance of the modulated signals, and in particular, the effectiveness of the prediction model in this study.

## 4. Summary and Conclusions

We investigated the ionospheric TEC time series to model and predict regular ionospheric variations and anomalies. This study focused on the TEC time series obtained from TEC GIM models with global coverage. We used the least-squares harmonic estimation for time series analysis. Multivariate and modulated least-squares spectra were estimated for the series to detect and subsequently model the regular and modulated dominant frequencies of the periodic patterns.

The multivariate analysis indicates that there are periods of $24/n$ hours; $365.25/n$ days ($n = 1, 2, \ldots$); 11 year periodic signals, as well as their higher harmonics; periods matching with the GPS draconitic year ($351.4/n$ days, $n = 1, 2, \ldots$); and periods around 27 days in the TEC time series. The modulated harmonic analysis results indicates that there are a set of peaks with periods of $(1 \pm j/365.25 = 1 \pm 0.0027)$ days ($j = 1, 2, \ldots, 5$) and $(1 \pm j/(11 \times 365.25) = 1 \pm 0.00025j)$ days ($j = 1, 2, 3$). The same situation holds also true for the higher harmonics of the diurnal signal.

Thereafter, a model consisting of a linear trend plus regular sinusoidal functions (pure signals), along with the modulated sinusoidal signals, was applied for studying anomalies and prediction. By investigation of the developed model, the following anomalies were detected:

- Semiannual anomaly: most of this effect occurs at low–mid latitude during the day, and the TEC value of the March equinox is significantly larger than that of the September equinox. The VTEC variation has a similar pattern to the September equinox in local time at different latitudes, but it is dissimilar for the June and the December solstices as an interchange between the southern and northern hemispheres at different latitudes.
- Winter anomaly: the intensity of the winter anomaly on high solar activity is more than that of low solar activity, and this anomaly is larger during the daytime than nighttime.

- Equatorial anomaly: this occurs between latitudes ~±20° peaking at ~±15 and from around 10:00 AM to the sunset.
- Evening anomaly: this has a clear peak around 10:00 PM, is likely to occur in August, and its highest value is observed in November. It also occurs approximately in low- and mid-latitudes, and can be observed in the mid-region in the summer.

By replacing the functional model, using only pure harmonic signals, with a functional model including both pure and modulated harmonic signals, we have improved the prediction of TEC values. The improvement of the RMSE value of the comparison between the prediction model and real data reaches to 3.2 TECu in some of the months. The modulated signals can improve a yearly average of RMSE value in the lower and higher solar activities by 0.7 TECu and 1.3 TECu, respectively. This indicates the importance of the modulated signals, and in particular, the effectiveness of our model to predict the TEC values.

The proposed model, consisting of a linear trend plus regular sinusoidal functions along with modulated sinusoidal functions, performs well for regional-scale GNSS observations or other techniques, e.g., using MU radar observations. However, this needs to be investigated and possibly improved in future studies. Short-term prediction for the regional models could also be another application of the proposed method. The irregular parts of the ionosphere could also be investigated in more detail in future work.

**Author Contributions:** Conceptualization, A.A.-S., M.R.; Data curation, M.R.; Formal analysis, M.R.; Funding acquisition, M.R. and H.N.; Investigation, M.R. and A.A.-S.; Methodology, A.A.-S.; Software, A.A.-S. and M.R.; Supervision, A.A.-S., H.N. and V.N.; Validation, M.R. and A.A.-S.; Visualization, M.R.; Writing–original draft, M.R.; Writing–review and editing, A.A.-S., H.N. and V.N. All authors have read and agreed to the published version of the manuscript.

**Funding:** This research was funded by the Iran Ministry of Science, Research and Technology and the APC was funded by Norwegian University of Science and Technology.

**Acknowledgments:** The authors would like to acknowledge the valuable comments of the editors and four anonymous reviewers, which significantly improved the presentation and quality of this paper. Also, the authors would like to thank the Jet Propulsion Laboratory (JPL) for data products which made this study possible.

**Conflicts of Interest:** The authors declare no conflict of interest.

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
