# Peer review of "Modeling and Prediction of Regular Ionospheric Variations and Deterministic Anomalies"

_remotesensing, doi:10.3390/rs12060936_

Round 1

Reviewer 1 Report

Review of “Modeling regular ionospheric variations using harmonic analysis of TEC time series”

The authors applied the Least square Harmonic Estimation to study the regular ionospheric anomalies as well as the TEC predictions. They found the regular anomalies like diurnal, annual, 11-year periodic signals, and even other periods. Then a model is derived based on these periods to predict the TEC values. The research is of interest to scholars in the field of the ionosphere.

Some minor comments are given:

Page6: The authors two cases of phi=0 and lamda=0 to analyze the periods. Did the authors try the other cases and they think there would be some differences? Page6: In Fig. 4, It can be noticed that there are some shifts between the periods and data. What could be the reasons? Page 5, equation 10, shall the RMS be squared? In Table 1, could the authors describe the table more in details, so that the readers do not need to refer back to the text to understand? The authors want to divide the table into low and high latitudes? What does the text in red mean? Page 14, there is “s” in the line before “Reference.”

Author Response

Dear Reviewer,

Hope you are fine and doing great,

We greatly appreciate the time and effort you took to read the manuscript in detail and for your detailed and constructive reviews that helped improve the quality of the paper and the shortcomings we had in the initial version. We considered all your points and comments as much as possible in the revised version.

please see the file uploaded below.

Best regards

Author Response

(The authors gave the same response as above.)

Reviewer 3 Report

The submitted study can be generally considered as extension of that in Amiri-Simkooei and Asgari (2012), however a round of English proofreading and text refining must be done and some additional unclear issues must be answered.

It is important to say what GIMs are used in the study, as they differ in sense of spatial resolution. Many models are based on harmonic expansion and limited to its low harmonics, e.g. 16 or so. The information in such models has spatial resolution even lower than their grid spacing, so also using 2.5 degree spacing in latitude by the authors seems aimless. Additionally, the number of details interesting from spectral time analysis point of view can be limited in these models. Please refer a little to this issue in the manuscript.

Is the presentation of the effect in Figs. 7-12 impossible from the original models, without spectral analysis and reverse modeling? Are the presented anomalies less visible in the original profiles from GIMs? Please refer also to this question.

Major comments:

The titles of two works are almost the same, but you state that you provide new studies. Please, change the title revealing the novelty.

Language is poor and incorrect in many places. I give some found examples below, but probably there are more. Please check carefully.

The annotations in Figs. 3 and 4 are unreadable. Please indicate new periodic signal transparently in the figures and text. This is the main novelty in comparison to the previous paper, the reader must to see it extremely clearly.

Please explain more how do you calculate power spectra from many latitude/longitude bands, i.e. how do you combine them? In other words how -87.5:2.5:87.5 refer to Fig 3? Also remember the above remark! If you work with model extended to n=m=15, only sampling at 180/15 band has true sense in my opinion.

You don’t use numbers of subfigures. There are some places, where they would be handy.

Sec 3.5. The modeling and prediction are the same from some point of view. It is better to distinguish between interpolation and extrapolation. Please be aware that interpolation is also a kind of prediction, and I’m sure you have to change the naming convention.

Example language mistakes:

The work needs language correction. I give some examples (but not always suggest the correct form) found during reading:

(Page/line)

1/16 -  draconian year – I saw the form: draconic year many times, but check, I’m not sure

1/16 - The results from the modulated harmonic analysis indicates that there exists - indicate that there exist

1/31 from about 60–2000km at the top - from about 60 km to  2000 km at the top

1/32 400km – 400 km

1/33 The main layers of that are known as D, E, F1 ,and F2 - Its main layers are D, E, F1 ,and F2

1/34 are hugely important, which has got considerable attention over the past decades - are very important, which focused a considerable attention over the past decades

1/44 The effects that seem not to be predictable - The effects that seem not predictable

2/46 day in winter compared to summer - day in winter compared to the summer

2/52 Though temperature fluctuations can be the reason for the observed - Although the temperature fluctuations can be the reason of the observed

61 The ionospheric effects on the GNSS signals - The ionospheric influence on the GNSS signals

67 as a method to analyze frequencies - as a method used to analyze the frequencies

78 and investigate model in local time - and investigate the model in local time

80 pure and modulate harmonic signals - pure and modulated harmonic signals

80 accomplished both in the low and high ionospheric – Do you mean low-frequency and high-frequency??

84 application on time series - application to ionospheric time series ?

Fig.2 – Compare two prediction – predictions

95 m-vector - m-element vector ? Check remaining cases of the same

101 Is this sentence finished in this line? Check

120 time series have been shown to consist of some - time series have been shown as consisting of some

131 and second predict TEC values by the identified model - and second, to predict TEC values with the identified model

134 Two scenarios used for - Two scenarios were used for ?

136 matrix A is made of the observation equations - matrix A is based on the observation equations

137 A least squares estimation of the pure and modulated signals – Least-squares estimation of pure and modulated signals

139 m-vector of observables - m-element vector of observables (check all cases)

141 A pure signal introduces two columns - The pure signal introduces two columns

143 time instants t1.….tm of which the time series observations - time instants t1.….tm, the time series of which …

146 with ? the total number of time - with ? being the total number of time

152  - absolutely rewrite this sentence

158 and are acquired with GNSS techniques - and the models are acquired with GNSS techniques?

166 spectrum for 71 TEC time series in ?=?° - spectrum for 71 TEC time series located at ?=?°…

175 the same kind of uncertainty, as above (and say how it is implemented to Fig 3 and 4)

Fig 3 and 4 – LARGE ANNOTATIONS OF PERIODS !

191, 196 – Do we know what is epsilon and delta here?

206 Figure 1 – Fig. 6 ??

210 Figure6 – Fig. 7 ??

209 The end of sentence is missed somewhere here, please check.

221 shows those – shows them? Consider please

224 on summer – in the summer

226 is larger in high solar activities than low solar activities - is larger in time of high solar activities than during low solar activities

227 is larger in daytime than in nighttime - is larger in the daytime than in the nighttime

228 occurs mostly in daytimes - occurs mostly by day

238 Evening anomaly – The evening anomaly

252 TEC values using the derived pure and modulated signals in this study - TEC values using pure and modulated signals derived in this study??

254 The first series concerns – word concerns looks strange in this context

257 The purpose is the TEC prediction for the different months of 2008 and 2013 – should be rewritten, also next sentence ending in 259

265 Figures 13 – Fig. 13 (a - …)

268 Title of the Table should give more info, RMS of what?, the same refers to Fig. 3

272 Which GIM model?

273 for the series to detect and hence model - for the series to detect and subsequently model??

283 By the investigation of developed model, the following anomalies were detected

And probably more, please refine the text

Author Response

(The authors gave the same response as above.)

Reviewer 4 Report

Dear authors,

this is the

Review report of the manuscript entitled

Modeling regular ionospheric variations using harmonic analysis of TEC time series

Mahmoud Rajabi, Alireza Amiri‐Simkooei, Hossein Nahavandchi, Vahab Nafisi

This report contains general comments and specific comments related to the submitted manuscript. Specific comments are given throughout the manuscript in for of highlights and comments (in attachment), while the general comments are provided below:

The English expression, grammar, and vocabulary through the manuscript must be revised thoroughly, preferably by a native English speaker or professional.

Abstract:

The abstract should be written in a more appropriate manner: introduction to the research topic, its relevance, etc. Please put the topic in a broader context.  

Introduction chapter:

This chapter should contain more Background elements, perhaps the background chapter should be provided as well since essential information is missing. The relation between the ionosphere and TEC should be provided more specifically and in detail, as well as an explanation of phenomena which occur: solar behaviour, storms, origins of stated variations, … all of these should be scientifically and technically supported.

The literature review is rather poor than satisfactory. A crucial, published (and also recent!)  achievements in the area are missing. Please stick to the topic and conduct a thorough review of other works related to TEC modelling in general, towards the specific problem which is elaborated here.

The model:

The proposed contribution is not clear. How this approach can be implemented further?

The conclusion chapter should be rewritten for the sake of proper expression and structure.  

There.

All the best

Author Response

(The authors gave the same response as above.)

Round 2

Reviewer 2 Report

Please, check the attached file.

Author Response

Dear Reviewer,

We thank you again for the time you spent to read the manuscript and for your comments. We hope the revised manuscript will meet your expectations.

Comment:

Page 1-2, line 45-46: In the opinion of this reviewer, the authors should better discuss the follow statement: “therefore, modeling and prediction of global VTEC are essential.

Reply:

We added some sentences, please see lines  44-47

Comment:

Page 4, line 166: The authors wrote “the coefficients ??1 and ??2 and ???, ?=1,2,3,4 are unknowns to 128 be estimated”. Please, check if is ?? or ??.

Reply:

Thank you for pointing this out.

Done.

Kind regards

Mahmoud Rajabi

Reviewer 4 Report

Dear authors, 

Thank you for your thorough response and appreciation of all of the remarks.

As by my side, the paper is ready to be considered for publication. 

With kindest regards 

Author Response

Dear Reviewer

We thank you again for the time you spent to read the manuscript and your positive feedback.

Best regards 

Mahmoud Rajabi